# How populist attitudes scales fail to capture support for populists in power

Sebastian Jungkunz[1,2,3]*, Robert A. Fahey[4]*, Airo Hino[5]

**1** Institute for Socio-Economics, University of Duisburg-Essen, Duisburg, Germany, **2** Chair of Political Sociology, University of Bamberg, Bamberg, Germany, **3** Center for Political Communication, Zeppelin University, Friedrichshafen, Germany, **4** Waseda Institute of Political Economy, Waseda University, Tokyo, Japan, **5** School of Political Science and Economics, Waseda University, Tokyo, Japan

* sebastian.jungkunz@uni-due.de (SJ); robfahey@aoni.waseda.jp (RAF)

**Data Availability Statement:** Replication material for all analyses can be accessed at the Harvard Dataverse (DOI: 10.7910/DVN/IRXXMO).

**Funding:** AH received funding from the Kakenhi Grant 20KK0026 by the Japan Society for the

## Abstract

Populist attitudes are generally measured in surveys through three necessary and non-compensatory elements of populism, namely anti-elitism, people-centrism, and Manicheanism. Using Comparative Study of Electoral Systems Module 5 (2016–2020) data for 30 countries, we evaluate whether this approach explains voting for populist parties across countries in Asia, Europe and the Americas. We show that the existing scales of populist attitudes effectively explain voting for populists in countries where populist leaders and parties are *in opposition* but fail to explain voting for populist parties in countries where they are *in power*. We argue that current approaches assume "the elite" to mean "politicians", thus failing to capture attitudes towards "non-political elites" often targeted by populists in office—in particular, journalists, academics/experts, bureaucrats, and corporate business leaders. The results reveal limits to the usefulness of existing survey batteries in cross-national studies of populism and emphasize the need to develop approaches that are more generalizable across political and national contexts.

## Introduction

With the rise of populist parties to power in many countries, it has become increasingly important to investigate not only the supply side of populism through the analyses of speeches and messages but also the demand and popular support for it. To do so, many (mostly European) scholars have developed several scales for measuring populist attitudes [1–6]. Unfortunately, the application of these scales has for the most part been limited to Europe and the Americas. With the exception of a single study that used convenience samples to compare all of these scales against each other [7], there has been little work on measuring populist attitudes cross-nationally in Africa, Asia or the Arab world. Module 5 of the Comparative Study of Electoral Systems (CSES) is the first study to have surveyed populist attitudes in most parts of the world through 2016–2020 using a representative sample of each national population. The data from CSES Module 5 therefore permit us to investigate the promises and pitfalls of using populist attitudes scales cross-nationally and under different political circumstances.

Promotion of Science (https://www.jsps.go.jp/english/e-grants/). SJ would like to acknowledge funding from the project "The influence of socio-economic problems on political integration" (PI: Paul Marx) funded by the North Rhine-Westphalian Ministry of Culture and Science. We further acknowledge support by the Open Access Publication Fund of the University of Duisburg-Essen. The funders had no role in study design, data collection and analysis, decision to publish, or preparation of the manuscript.

**Competing interests:** The authors have declared that no competing interests exist.

We argue that the application of populist attitudes scales across time and region can be problematic for two reasons. First, many scales include references to a vaguely specified "elite" which can be interpreted quite differently by respondents in different countries or even within the same country at different times. As we discuss below in detail, anti-elitism is conceived as an essential dimension of populist attitudes, meaning this limitation can be crucial. Secondly, in cases where the elite is actually specified, almost all existing scales refer exclusively to the *political* elite—parties, politicians or government. The assumption that these are the elites perceived by populist voters as acting against the interests of the common people, however, does not hold when populist parties and leaders have been elected into power. One possibility is that the successful election of populist politicians would satisfy the demands which had led voters to hold populist attitudes, thus resulting in a decline in those attitudes (in which case populism would be acting as a "thermostatic" attitude [8], signalling a demand for change and then dissipating once change has been achieved). The other possibility that we highlight here is that populists elected to power and their supporters simply shift the targets of their anti-elite sentiments to other *non-political* elite groups such as journalists, academics/experts, bureaucrats (as exemplified by the so-called "deep state"), and corporate business leaders. In these cases, the narrow focus on political actors found in existing ways of gauging populism would fail to measure these attitudes due to the change in how the "elite" is defined and framed, even though the broader nature of the attitudes remains unchanged.

Our analysis of the CSES data finds that such scales work as expected in many countries, but that there is indeed a disconnect between populist attitudes and voting behavior in countries where populist parties are actually in power—in these countries, higher populist attitudes as measured on the CSES scale actually decrease the likelihood of voting for populist parties. Robustness checks with five other populist attitudes scales in Greece and Japan indicated similar findings. The results highlight the existence of multiple varieties of populism in different political and social contexts, suggesting that existing definitions and methods may be unsuitable for cross-national comparisons.

## Measuring populist attitudes cross-nationally

The *ideational approach* considers populism to be a thin-centered ideology which is grounded in a Manichean worldview of good against evil, in which evil elites conspire against the good people [9–11]. Ordinary citizens, characterized as good, pure, and homogeneous [1, 12], are considered to constitute a silent majority whose *volonté générale* (general will) should rightly be the base of political decision-making, but is instead suppressed by a corrupt elite [13]. The elite is often only vaguely specified and can refer to political or economic actors, state employees, intellectuals, journalists or the media in general [12]. Populism results when these three elements—people-centrism, anti-elitism and a Manichean worldview—coincide. Populist attitudes are widespread among ordinary citizens in most if not all countries, regardless of the actual presence of populist political actors [14, 15].

In general, cross-national measurement of attitudes can be subject to a variety of errors that may be grouped into three categories: construct, method, and item bias [16–18]. Construct bias occurs when a construct has a different meaning in one country compared to another. Method bias occurs when there are differences in the data collection procedures. For instance, cross-national studies often vary in their sampling design [19], unit and item nonresponse patterns [20], or in the administration of questionnaires [21]. Finally, item bias results from a different functioning of individual items in questionnaires. Often, this is the result of poor translations from the original source questionnaire or the inclusion of country- or culture-specific terms that are understood differently across countries.

All of these can have substantial consequences for the cross-national measurement of populist attitudes. One factor related to the above construct and item biases seems to be of particular importance to this kind of measurement: references to the "elite" in populist attitudes questions can leave substantial scope for variation in what and whom respondents actually consider to be the elite. For instance, most populist attitudes batteries contain questions like whether "The political differences between the elite and the people are larger than the differences among the people". As the elite is often only vaguely specified even by populists themselves, the concept can vary widely both between and within countries. In situations where populist opposition parties are challenging more established political parties—as was the case in most contexts in which existing populist attitude scales were developed and tested—their rhetoric generally targets political actors. The problem arises once they occupy positions of political and governmental power themselves. We might expect that the populist attitudes of their voters had been satisfied and extinguished by their election, in accordance with the thermostatic view of populist attitudes. Or alternatively, as we try to illuminate, their rhetoric may not lose its populist nature and instead shifts its targets to other elite groups.

The examples of populist rhetoric targeting other elite groups abound. In the United States, the anti-elite rhetoric of Donald Trump targeted the Washington DC bureaucratic establishment ("the swamp", or the "deep state") and on the "lying media" (and shifted back somewhat to more traditional political targets following his departure from office at the beginning of 2021). In Hungary, Viktor Orbán and the ruling FIDESZ party have often attacked academics, and certain journalists and media, labelling them as a liberal elite that conspires against him [22, 23], while in Poland, Andrzej Duda of the ruling Law and Justice Party implied during his 2020 presidential election campaign that sexual minorities were a conspiring elite with a "destructive" ideology. In Japan, regional populist leader Hashimoto Toru's anti-elite rhetoric focused on the claimed corruption of local bureaucrats along with attacks on labor unions and school teachers [24–26]. These changes in the targets of populist rhetoric track with a changing perception of the elite among supporters of elected populist parties: evidence from Bolivia and Ecuador further shows that populist supporters no longer perceived the federal government to be part of the political elite once their leader had been elected into power [27]. In India, the populist conception of "the people" who are oppressed by the "elite" also differs from other nations and continues to evolve as Narendra Modi tries to redefine "the people" to refer exclusively to Hindus [28]. Thus, the perception of what constitutes the "evil elite" can differ significantly from country to country, and change when populist parties come into power.

Survey questions designed to measure anti-elite attitudes often fall into one of two broad camps, either leaving it entirely up to the respondent to interpret "the elite" as they see fit, or, perhaps more commonly, asking specifically about attitudes towards politicians and the government—in effect narrowing the definition of the concept to *political* elites. Each of these approaches has major potential pitfalls, especially in comparative research, as respondents in different contexts may have very different interpretations of anti-elitism, including the possibility that some individuals may hold strongly populist attitudes whose anti-elite component is not especially focused on politicians or government. This creates the risk for survey researchers and comparativists of comparing apples to oranges [29], as it is impossible to tell whether a difference between two groups is the result of a true difference or merely a different understanding of survey items [30]. Unfortunately, it has been shown that a variety of constructs are not fully invariant across nations and cultures, e.g. attitudes towards democracy [31], left-right orientation [32], trust in government [33], values [34] or xenophobia [35]. Such measurement problems are not confined to ordinary survey respondents, but are also found among experts in the field, as has been shown in the case of the Perceptions of Electoral Integrity (PEI) data [36].

The questions on "anti-elite" attitudes in the CSES Module 5 data we use in this study fall into the latter of the two groups—they assume *a priori* that respondents' anti-elite attitudes will be targeted at political and governmental figures. For instance, the surveys asked whether "Most politicians do not care about the people" or whether "Most politicians care only about the interests of the rich and powerful". In a context where populist attitudes are strongly focused on a different "elite" group—such as journalists, business leaders, or an ethnic or religious minority group, for example—these questions would fail to capture the anti-elite dimension entirely, since they simply do not ask about anti-elite attitudes targeting any group other than politicians. As the dimensions of populism are generally considered to be a non-compensatory construct (see [37])—meaning that all of the dimensions must be present, and a high score in one dimension cannot compensate for the absence of another dimension—this failure to measure the anti-elite dimension would result in the respondent being labelled as non-populist overall. Hypothetically, one can imagine a situation where the supporters of a populist challenger party (which has campaigned on a platform of strongly anti-government, anti-politician populist rhetoric) score very highly in a survey battery on populist attitudes conducted prior to an election—but if the same voters were to be re-surveyed some time after their preferred populist party had won that election and taken power, subsequently shifting its anti-elite rhetoric to target bureaucrats, the media, labour unions or some other group, the voters' attitudes would no longer register as populist due to the failure to measure their newly refocused anti-elite views. If this data were instead to be a snapshot (like the CSES data), it would appear, paradoxically, that a populist party had been voted into power by non-populist voters—in fact, the satisfaction of those voters with their party now being in power could create an unexpected negative relationship between populist sentiment among voters and likelihood of voting for the populist party. Of course, if the survey were instead to ask about voters' anti-elite attitudes towards the specific targets of the party's new anti-elite rhetoric, quite a different measurement would result, one that would be more in line with the basic expectation that populist voters vote for populist parties.

While for the purposes of this study we treat the issues outlined above as a measurement error, this relies upon two assumptions: firstly, that populist attitudes are not purely thermostatic and thus do not dissipate once a populist party or leader has been elected to power, and secondly, that we are working with a broad conception of populism in which anti-elite attitudes targeting non-political actors satisfies the anti-elitism component of the definition, as distinct from a strictly political conception of populism in which only anti-politician or anti-government attitudes satisfy this condition. To the first point, as we have argued above, the continued use of populist rhetoric (albeit with new, non-political targets) by prominent populists in power such as Donald Trump and Viktor Orbán implies that their voters continue to hold populist attitudes, and thus to be responsive to populist rhetoric, despite the electoral success of their chosen candidates, meaning that the thermostatic view of populist attitudes is not fully applicable—or rather, that populist attitudes may indeed be thermostatic to some extent, but that they are not straightforwardly satisfied just by the election of populist parties and leaders. This is in line with the argument that—like some of their component aspects, such as the Manichaean worldview—populist attitudes are quite a fundamental way of viewing society in general and political representation in particular [38], and as such we would not expect people's populist attitudes to change quite so rapidly or so easily. To the second point, there may certainly be occasions in which using a tightly-defined conception of populism for which only anti-elite attitudes targeting political actors qualify is appropriate and useful. However, we argue that if this political anti-elitism is merely replaced with equivalent hostility towards other actors or sections of society following the election of populist parties, it is in most cases too narrow an interpretation to simply say that populist attitudes have disappeared.

These re-targeted sentiments may not be political populism by some definitions—a discussion which is beyond the scope of this paper—but the most widely accepted definitions of populism do not specify that the targets of anti-elite sentiment must be political actors, and the attacks on journalists, academics, business leaders, bureaucrats and other non-explicitly political targets by populists in government show that this phenomenon is deserving of exploration.

This study thus aims to test the external validity of the measurement of populist attitudes across cultures and contexts, including those in which populist parties are in power (or in which populist rhetoric has been embraced by mainstream parties) rather than being challenger parties. To test the external validity of these items, it is necessary to check how strongly they correlate with related constructs [7, 39]. In theory, we would expect populist attitudes to predict vote choice for populist parties in each country; populist attitudes are of course a much broader and more widely applicable concept than simply being a predictor of voting behavior, but voting for populist parties is one of the directly measurable behaviors we can reasonably expect to be correlated with such attitudes. Thus, we should expect that a high level of populist attitudes increases the likelihood of voting for populist parties, as has been shown elsewhere [40]—but as in the hypothetical example given above, we hypothesize that this connection will disappear or reverse in situations where the populist party is in power. To investigate this proposition, we first assess the model fit in each country and test whether the scale fits equally well in nations where populists are in power. We then look at how well this measurement of populist attitudes actually predicts vote choice for populist parties. If the measurement of populist attitudes works equally well across countries (including those where populists are in power), we should consistently find positive relationships with populist vote choice. If, however, the survey items fail to capture anti-elite sentiments in situations where populists are in power, as we hypothesize, this would lead to a failure to correctly identify voters with populist attitudes in those countries (specifically, the model would misidentify voters for populist parties in power as being non-populist). We would therefore expect to find diverging patterns for countries where populist parties are in power.

## Data & operationalization

In order to investigate the structure and predictive potential of populist attitudes batteries we use data from the *Advanced Release 3* of CSES Module 5, which surveys populist attitudes in 28 countries [41]. We add to these the data from election surveys in Japan [42], the Netherlands [43], and the US (in 2020) [44], as they have already been released but not yet added to the CSES Advanced Release. In sum, the data include 55,515 respondents in 30 countries for 34 elections. An overview can be found in S1 Table in S1 File. All replication materials can be accessed via the Harvard Dataverse [45].

Unfortunately, operationalizing populist attitudes in the CSES Module 5 is not straightforward, as its questions are not drawn from one of the major previously available scales for populist attitudes. Instead, it proposes a multidimensional concept which measures three "core themes" of populism: attitudes towards political elites, attitudes towards representative democracy and majority rule, and attitudes towards out-groups. Although the CSES Planning Committee refers to the work of Mudde ([46], 5), it is less clear which of these themes are considered necessary elements of populism and which are related constructs. For instance, attitudes towards out-groups are not considered to be a central element by all scholars [47, 48]. Moreover, the dimension is measured with items which ask about respondents' attitudes towards immigrants and the importance of nationality and national customs and traditions. Such an operationalization may be applied if researchers want to measure *right-wing* populist

**Table 1. Populist attitudes items in the CSES Module 5.**

| Item | Wording | Castanho Silva et al. (2020) | Wuttke et al. (2020) |
|---|---|---|---|
| E3004_1 | Q04a. What people call compromise in politics is really just selling out on one's principles. | Challenges to representative democracy (Manichean Worldview) | Challenges to representative democracy |
| E3004_2 | Q04b. Most politicians do not care about the people. | Anti-Elitism | Anti-Elitism |
| E3004_3 | Q04c. Most politicians are trustworthy. | Anti-Elitism | Anti-Elitism |
| E3004_4 | Q04d. Politicians are the main problem in [COUNTRY]. | Anti-Elitism | Anti-Elitism |
| E3004_6 | Q04f. The people, and not politicians, should make our most important policy decisions. | People-Centrism | Challenges to representative democracy |
| E3004_7 | Q04g. Most politicians care only about the interests of the rich and powerful. | Anti-Elitism | – |
| E3005_2 | Q05b. The will of the majority should always prevail, even over the rights of minorities. | – | Challenges to representative democracy |
| E3007 | Q07. How widespread do you think corruption such as bribe taking is among politicians in [COUNTRY]? | – | Anti-Elitism |

Castanho Silva et al. used a one-dimensional operationalization and did not specify specific sub-dimensions for the CSES items. We added these to indicate how a possible three-dimensional operationalization might look.

attitudes, but it is definitely too specific as a general, non-ideological operationalization of the thin-ideology that populism is regarded to be [37].

To address these issues, we decided to test multiple operationalizations of populist attitudes in the CSES, based on two previous works (see Table 1). The study by Castanho Silva et al. uses a selection of the CSES items in order to compare a wide range of currently available populist attitudes scales [7]. These items can basically be grouped into the traditional dimensions of the work by Mudde, i.e. anti-elitism, people-centrism and a challenge to representative democracy (which shares the anti-pluralist component of a Manichean worldview). However, item E3004_1 does not adequately fit the dimensions of a Manichean worldview. Although it includes a sense of anti-pluralism, it does not properly reflect the moral struggle between two opposing groups. We therefore labelled it slightly differently. In addition to that, we use the operationalization by Wuttke et al. which is closer to the original proposal by the CSES Planning Committee, excluding the out-group and minority dimension [37]. All items were measured on five-point Likert scales. However, item E3004_1 is worded positively ("In a democracy it is important to seek compromise among different viewpoints") in Greece, Hong Kong, Ireland, South Korea and Taiwan (2016), as the data was part of a pre-test of the CSES. However, corruption perceptions like in item E3007 are generally not considered to be part of populist attitudes [49], as populists do not actually try to fight corruption in the sense of tackling the misuse of political power for personal gains [49]. Rather, they consider the elites to be corrupted by the mere involvement in politics [50]. As this differentiation is not adequately reflected by item E3007 ("How widespread do you think corruption such as bribe taking is among politicians in [COUNTRY]?"), we provide additional analyses excluding the item in the (see S7 and S8 Figs in S1 File). The results indicate, however, that removing the item produces nearly identical results.

Just as it is difficult to agree on which items to use, it is also questionable which method of aggregation is best to capture the concept of populist attitudes. For most populist attitudes scales, scholars use a simple mean or additive index in which the mean is taken across all items. This works well for compensatory concepts, i.e. if all items are considered to be equally important. However, this can become complicated if the list of items is unbalanced, i.e. there are more items that belong to one dimension as compared to another one. As we can see in

Table 1 this applies to both operationalizations. More advanced or weighted versions of measurement therefore try to aggregate dimensions and then calculate means across dimensions. Although technically superior, this logic also applies to the use of structural equation modelling and second-order confirmatory factor analyses. Unfortunately, such approaches are not ideal for measuring non-compensatory concepts, i.e. when a lack in one dimension cannot be made up for with high values on another one. As populism is considered to be the intersection of anti-elitism *and* people-centrism *and* Manichean worldview at the same time, it is considered a non-compensatory concept. To score high on a populist attitudes variable, respondents would thus need to have high values on all three dimensions. Wuttke et al. therefore proposed a different approach based on the work of Goertz [51, 52], which takes the minimum value of all subdimensions.

In sum, we thus use two different versions of the CSES items along with three methods of aggregation. For the more traditional version used by Castanho Silva et al. we use structural equation modelling (a) to form a one-dimensional scale and (b) to form a three-dimensional scale. For the three dimensional version by Castanho Silva et al. and the two-dimensional version by Wuttke et al. we further use the favored method by Wuttke et al., taking the minimum value of both dimensions (c & d). Accordingly, we first standardized all items by country to a mean of 0 and a standard deviation of 1. We then calculated the average values of each dimension and recoded the dimensions to a scale from 0 to 1. To build the Goertzian version of the scales we then took the minimum values across the two, respectively three dimensions ($Goertz := min|Dimension_1, ..., Dimension_n|$).

Finally, we also ran additional robustness checks to test whether our results are subject to the CSES items in particular, or whether they relate to most of the existing scales in the field to this day. To do so, we conducted an online survey in Japan—a complex case in terms of populist politics—in 2019, which included the Castanho Silva et al. and Schulz et al. populist attitudes scales. Furthermore, we used the 2016 Greek data from Castanho Silva et al. [2] which contained the CSES items along with five other populist attitudes scales ([1–3, 5, 6]) to predict vote choice for SYRIZA, a clear-cut populist party in power at that time. Each of these scales operationalizes populism in a somewhat different way: both the Akkerman et al. scale and the Elchardus and Spruyt scale are exclusively focused on questions related to political representation (the latter leaning more heavily on negative views of experts, while the former is more focused on popular sovereignty), while the Castanho Silva et al. scale and the Schulz et al. scale both attempt to faithfully replicate a three-dimensional definition (people-centrism, anti-elitism, and Manichaean outlook for Castanho Silva et al., and anti-elitism, popular sovereignty, and homogeneity of the people for Schulz et al.). The Stanley scale, finally, consists mostly of reverse-coded items (positive statements about pluralism, the importance of expertise, and so on) but, like the Akkerman et al. and Elchardus and Spruyt scales, is largely focused on statements related to political representation and democracy. Despite these significant conceptual differences among the scales, the results in the Greek case are basically in line with the findings presented in the text (see Appendix B in S1 File), supporting the argument that they all share the same fundamental focus on political elites to the exclusion of other types of elite actors.

## Psychometric assessment

In a first step we investigate how well the proposed operationalizations fit the data in each country in the CSES. For all analyses we used *Mplus* version 8 [53] through the *R* [54] package *MplusAutomation* [55]. All data handling, preparation and visualization were carried out in *R* using a variety of different packages [56–64]. As shown in Table 2, the one-dimensional CSES scale used by Castanho Silva et al. has a medium to good fit in most countries according to

**Table 2. Confirmatory factor analysis models Castanho Silva et al. version (one dimension).**

| Country | N | RMSEA | SRMR | CFI | Avg. Loading | Min. Loading | Lowest Loading |
|---|---|---|---|---|---|---|---|
| Austria | 1020 | 0.063 | 0.024 | 0.982 | 0.671 | 0.436 | E3004_6 |
| Australia | 1666 | 0.062 | 0.026 | 0.978 | 0.633 | 0.524 | E3004_6 |
| Belgium (Flanders) | 922 | 0.095 | 0.031 | 0.960 | 0.669 | 0.528 | E3004_3 |
| Belgium (Wallonia) | 583 | 0.096 | 0.034 | 0.951 | 0.631 | 0.450 | E3004_1 |
| Brazil | 1933 | 0.035 | 0.021 | 0.977 | 0.432 | 0.199 | E3004_3 |
| Canada | 1974 | 0.048 | 0.019 | 0.982 | 0.574 | 0.438 | E3004_6 |
| Chile | 1324 | 0.073 | 0.038 | 0.914 | 0.428 | 0.138 | E3004_1 |
| Costa Rica | 1030 | 0.041 | 0.023 | 0.974 | 0.449 | 0.251 | E3004_1 |
| Finland | 902 | 0.069 | 0.027 | 0.977 | 0.655 | 0.481 | E3004_3 |
| France | 1286 | 0.072 | 0.029 | 0.968 | 0.606 | 0.386 | E3004_1 |
| Germany | 1659 | 0.025 | 0.011 | 0.997 | 0.689 | 0.540 | E3004_3 |
| Great Britain | 781 | 0.099 | 0.036 | 0.947 | 0.630 | 0.447 | E3004_3 |
| Greece[1] | 717 | 0.080 | 0.038 | 0.923 | 0.462 | 0.125 | E3004_1 |
| Hong Kong[1] | 722 | 0.066 | 0.038 | 0.811 | 0.321 | 0.097 | E3004_1 |
| Hungary | 832 | 0.077 | 0.032 | 0.961 | 0.565 | 0.208 | E3004_1 |
| Iceland 2016 | 848 | 0.055 | 0.023 | 0.983 | 0.624 | 0.395 | E3004_6 |
| Iceland 2017 | 1381 | 0.044 | 0.018 | 0.989 | 0.614 | 0.372 | E3004_6 |
| Ireland[1] | 827 | 0.034 | 0.017 | 0.992 | 0.531 | 0.034 | E3004_1 |
| Italy | 1260 | 0.079 | 0.036 | 0.937 | 0.520 | 0.280 | E3004_1 |
| Japan | 1352 | 0.057 | 0.027 | 0.959 | 0.464 | 0.112 | E3004_1 |
| Lithuania | 1008 | 0.101 | 0.045 | 0.916 | 0.527 | 0.181 | E3004_1 |
| Montenegro | 806 | 0.062 | 0.026 | 0.978 | 0.588 | 0.158 | E3004_3 |
| Netherlands | 2355 | 0.074 | 0.022 | 0.979 | 0.706 | 0.585 | E3004_3 |
| New Zealand | 1290 | 0.064 | 0.025 | 0.977 | 0.620 | 0.488 | E3004_3 |
| Norway | 1583 | 0.052 | 0.021 | 0.984 | 0.629 | 0.506 | E3004_6 |
| Portugal | 1152 | 0.058 | 0.028 | 0.968 | 0.522 | 0.245 | E3004_1 |
| South Korea[1] | 1179 | 0.047 | 0.028 | 0.944 | 0.379 | 0.128 | E3004_1 |
| Sweden | 3170 | 0.077 | 0.026 | 0.975 | 0.620 | 0.422 | E3004_3 |
| Switzerland | 3826 | 0.034 | 0.015 | 0.992 | 0.590 | 0.383 | E3004_3 |
| Taiwan 2016[1] | 1248 | 0.058 | 0.032 | 0.936 | 0.407 | 0.069 | E3004_1 |
| Taiwan 2020 | 1350 | 0.042 | 0.024 | 0.970 | 0.424 | 0.099 | E3004_1 |
| Turkey | 912 | 0.064 | 0.030 | 0.955 | 0.485 | 0.135 | E3004_1 |
| United States 2016 | 3481 | 0.061 | 0.027 | 0.966 | 0.540 | 0.372 | E3004_6 |
| United States 2020 | 6734 | 0.066 | 0.026 | 0.966 | 0.566 | 0.393 | E3004_6 |

[1] Part of the pre-test with reversed E3004_1 item. The Swedish data does not contain E3004_1. Shown are standardized loadings.

common goodness-of-fit criteria [65]. Only in Hong Kong and Lithuania does the scale fit rather poorly. Overall, there does not seem to be a strong cross-cultural difference in model fit, i.e. we tentatively argue that populism, as it was modeled here, fits the data well. However, there are substantial differences in the average loadings across countries, as the average loadings in Asian and Latin American countries are substantially lower than in European countries or the United States. Thus, it seems that the concept works better in those countries where it was originally developed. It is therefore questionable whether populism can be measured in a fully invariant manner across cultures. Future research could test for this once more data is available. Finally, we see that the minimum loading in most countries pertains to the same item (E3004_1), "What people call compromise in politics is really just selling out on one's

principles," which has a low loading in many countries regardless of whether the item has a positive (as in the CSES pre-test) or a negative wording. In a three-dimensional measurement model this item would constitute the sole measurement of Manicheanism in the scale, but it is questionable whether it adequately captures the Manicheanism dimension (we note that Wuttke et al. classifies it instead as one of three items measuring "challenges to representative democracy"). Aside from the consistently low loading raising concerns about how well this item fits with the scale, the wording itself is focused narrowly on political compromise, which may fail to capture the "good vs. evil" mindset of some respondents. This potential problem is consistent with the larger issue we believe to exist with the items measuring anti-elite attitudes—in each case, the wording narrows the concept by specifically locating it in the realm of politics and politicians. In the absence of an alternative item to measure Manicheanism, we use it here despite the low loading, but note that this is another dimension of populist measurement which researchers should be cautious about using without due attention to precisely what is being asked and measured, and how it differs from the original concept outlined in the literature.

We next tested the same version of the scale and constructed a three-dimensional model as outlined in the third column of Table 1. The advantage of this approach is that equal weight is put on all three dimensions, which then form an overarching second-order factor of populist attitudes. Accordingly, items E3004_2, E3004_3, E3004_4 and E3004_7 constitute the anti-elitism factor, whereas people-centrism (E3004_6) and Manichean worldview (E3004_1) were single indicator latent variables. Technically these are estimated by setting the factor loading of E3004_7 and E3004_1 to 1 and the error variance to a non-zero value in order to account for the non-perfect reliability [66]. The value is set as: $\delta_x = Var(x) * (1 - \rho)$ ([67], 122), using scale reliability $\rho$ from an earlier project which measured these dimensions with multiple items (see [2]), giving scale reliability for people-centrism and Manichean worldview of 0.765 and 0.790 respectively. The results of the three-dimensional model in S2 Table in S1 File show similar model fit results to those presented in the text. Of course we cannot compare the size of the loadings, as those are much higher by nature when using single-item latent variables.

On a side note, we also estimated the model fit of the CSES populist items used by Wuttke et al. for comparison (see S3 Table in S1 File). All countries from the pre-test, Sweden and Taiwan were excluded, as they did not ask item E3005_2. Although this is, strictly speaking, not how Wuttke et al. intended to operationalize the items and how we use it in the rest of the study, we believe that this merits some discussion as it is the first time that the items are tested in such a large cross-national comparison. In general, the model seems to fit the data well in most countries. Only Lithuania and to some degree Great Britain and Turkey show a somewhat lower fit. That said, we have to recognize though that one item experiences substantially lower loadings in almost all countries. E3005_2 which asks whether "The will of the majority should always prevail, even over the rights of minorities", does not seem to be related to the concept of populist attitudes quite that much. In fact, one could argue that the item probably taps into related right-wing concepts, but not the general thin ideology of populism.

## External validity: Predicting vote choice

To test the external validity of populist attitudes scales we estimated the effect of populist attitudes on vote choice. The CSES distinguished between vote choice in parliamentary and presidential elections. We used vote choice in parliamentary elections except in Brazil, Chile, Costa Rica, France, Turkey, Taiwan, and the US, where we used presidential elections. For the Castanho Silva et al. version of the CSES items we ran structural equation models (SEM) with a multinomial dependent variable. Fig 1 gives a conceptual overview of the model using the one-

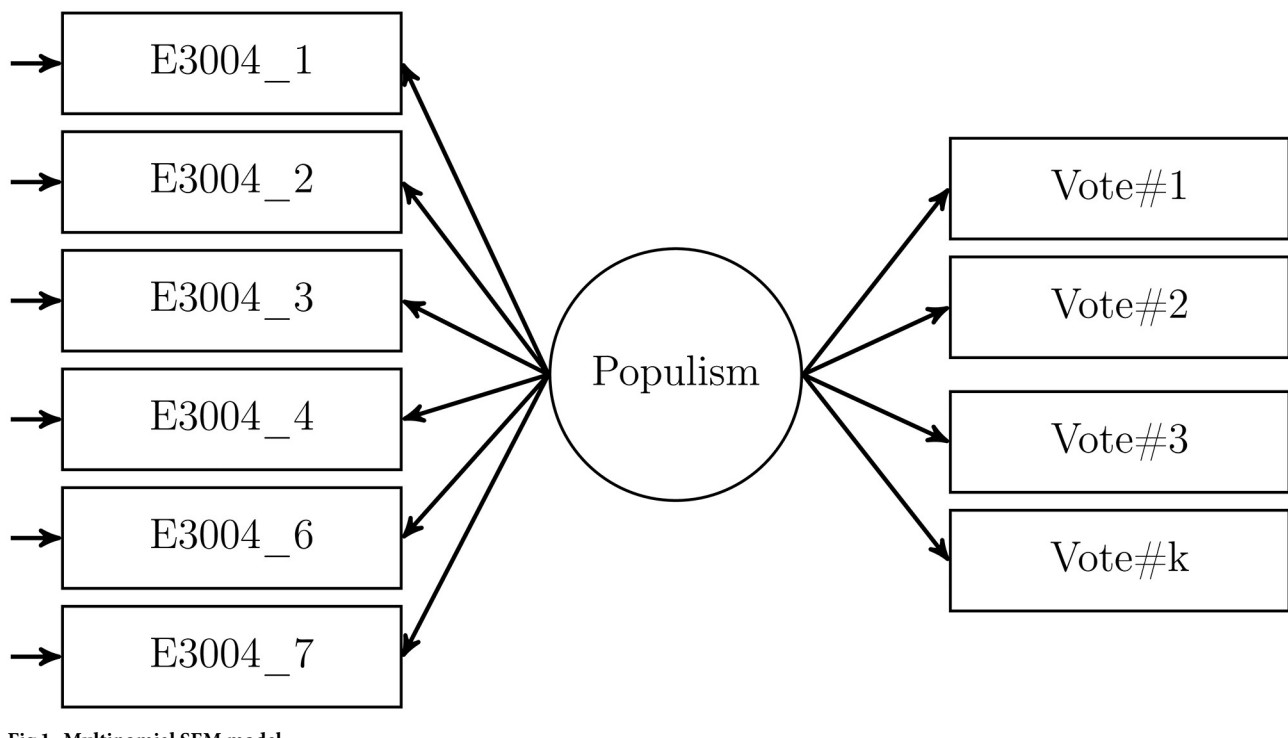

**Fig 1. Multinomial SEM model.**

dimensional operationalization. As such models can become difficult to estimate if there are only few cases in some of the categories of the dependent variable, we recoded the country-specific variables for voting behavior to include only those respondents who voted for a party which was elected to parliament in the election surveyed. We also added all those respondents who indicated that they did not vote in the election. Due to the very large number of respondents in Hong Kong (n = 296) and Portugal (n = 281) who refused to answer the question about vote choice, we also included "refused" as an additional category in both countries.

The predictions of populist vote choice using the one-dimensional Castanho Silva et al. version of the CSES items are visualized in Figs 2 and 3. Parties generally considered to be populist are highlighted in red and labels of interesting other parties are added to the individual plots. We coded populist parties along the schema provided by the *PopuList* [68]. For Asia and Latin America we used the recommendations of local experts and prior literature. Parties with a low number of cases were omitted for the sake of readability. If the CSES populist attitudes scale has a high external validity, we should find an upward curve for all populist parties and a downward slope for all non-populist parties. Of course, the relationship is not perfect, as populist attitudes are held across the political spectrum (see [14, 15]). But at least we should find trends in that direction. From these results we can conclude that populist vote prediction works well in most countries, with a high degree of populist attitudes coinciding with a higher probability to vote for a populist party. The absolute probability to vote for a party is of course subject to the number of respondents who voted for the party in the first place. Thus Ireland's Sinn Fein, for instance, shows a low overall probability, though it increases slightly as populist attitudes increase. On the other hand, we find mixed effects in countries that were considered to lack a populist party at the election surveyed. Whereas Labour parties in the Commonwealth seem to attract voters with high populist attitudes (Australia, New Zealand and UK), other

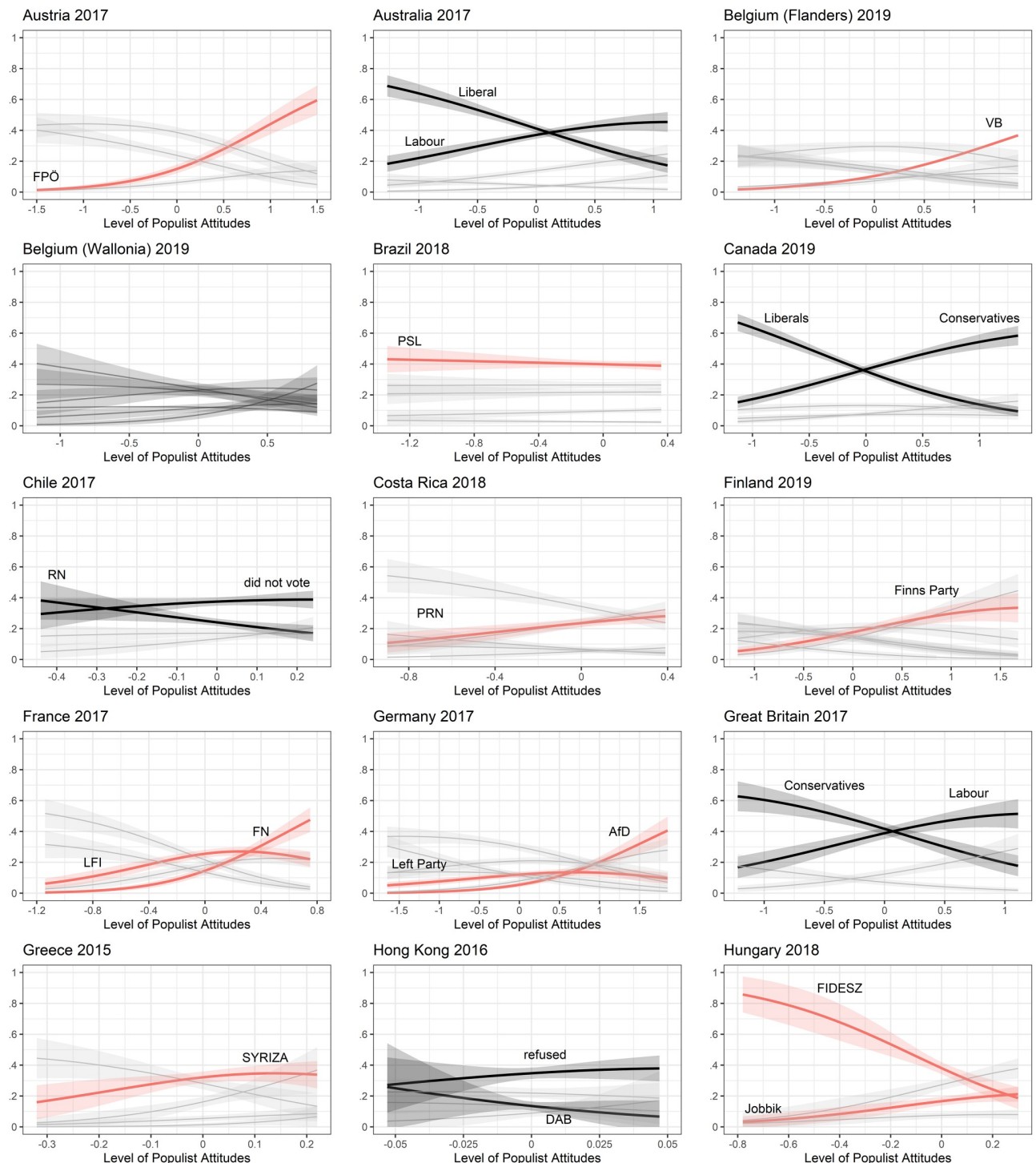

**Fig 2. Predictions of populist vote across countries I.** Predictions of vote choice are based on multinomial structural equation models with 95% confidence intervals.

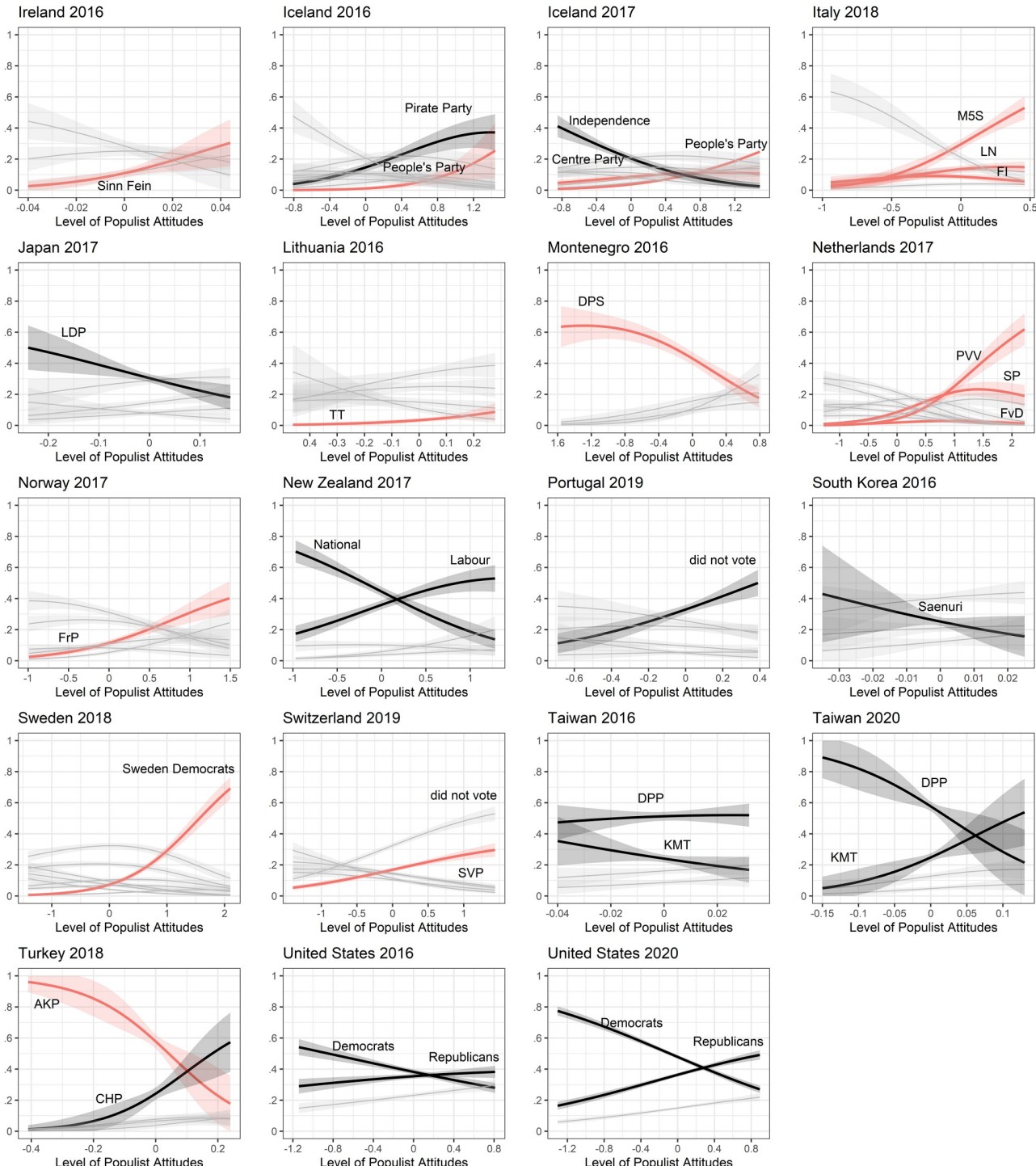

**Fig 3. Predictions of populist vote across countries II.** Predictions of vote choice are based on multinomial structural equation models with 95% confidence intervals.

countries experienced only small positive effects of populist attitudes on vote choice for some parties (Chile, Hong Kong, Taiwan and the US in 2016).

That said, the far more interesting results here are the cases in which we see the opposite relationship. In Hungary, Montenegro and Turkey, higher populist attitudes actually *decrease* the probability to vote for the respective populist party in each country. Voting for Japan's Liberal Democratic Party (LDP) and South Korea's Saenuri Party, both of which are argued by some scholars (albeit not uncontroversially) to be examples of mainstream conservative parties that have adopted a subset of populist strategies, also have negative relationships with populist attitudes ([25, 26, 69]; see also Appendix B in S1 File). These seemingly paradoxical results are in line with our expectations outlined above: in all of these countries the populist party was the incumbent ruling party at the surveyed election. The CSES populist attitudes scale measures the anti-elite dimension of populism through items that refer explicitly to *politicians* as being corrupt or not trustworthy—and in these cases, political and governmental power was associated with the preferred party or leaders of populist voters. This is not just a problem of the CSES populist attitudes items, but one common to almost all such scales. The supporters of populist parties in power are unlikely to agree with statements about the corruption of political leaders; having installed their preferred leaders, they no longer consider political leaders to be the major problem of the country or to be conspiring against the good and honest people (see also [27]). In turn, populist parties who win power often change their communication to focus condemnation on other, non-political elites, e.g. academics, bureaucrats, journalists or the media. Consequently, the scale fails to correctly identify populist attitudes, and thus to predict populist vote choice, in these cases where populists were in power.

One notable exception to this pattern is the United States, where the incumbency of President Donald Trump at the 2020 election does not appear to have impacted the scale's measurement of populist attitudes among his voters—in fact, the populist attitudes of Republican voters in 2020 are more clearly delineated than they were in 2016. This does not, however, contradict the overall argument regarding the over-specification of the anti-elite dimension, since Trump had an unusually fractious and combative relationship with his own political party, to the point of overtly attacking senior Republicans whom he perceived as disloyal or weak. This rhetoric stepped up during his reelection campaign in 2020 and intensified further during his subsequent attempts to discredit the election results, to the point where the Trump supporters who attacked Congress in January 2021 were heard chanting slogans calling for the execution of Trump's own Vice President, veteran Republican Mike Pence. Given these events, we would not reasonably expect to see Trump's voters following the trend observed elsewhere of refocusing their anti-elite sentiments away from politicians.

These results are confirmed using various other operationalizations of populist attitudes. Fig 4 provides an overview for all countries where populist parties were in power prior to the election at which the survey took place (see S1-S6 Figs in S1 File for all countries). As a result, we find that the negative relationship between populist attitudes and vote choice for populist parties holds in all countries where populist parties were in power prior to the election regardless of operationalization procedure. Using the three-dimensional version which puts equal weight on all three dimensions (anti-elitism, people-centrism and Manichaenism) yields more or less similar effects in Hungary, Montenegro and Turkey. Somewhat less clear-cut but still exhibiting a downward trend are the results from the Goertzian operationalization of the Castanho Silva et al. items of the CSES scale. Especially in Hungary and Turkey the effects are smaller when we use the minimum value of the three dimensions. Moreover, if we use the same method of operationalization for the suggested items by Wuttke et al., the effects again indicate a strong negative relationship between populist attitudes and populist vote choice. The effects for populist attitudes on populist vote choice are very similar in all other countries

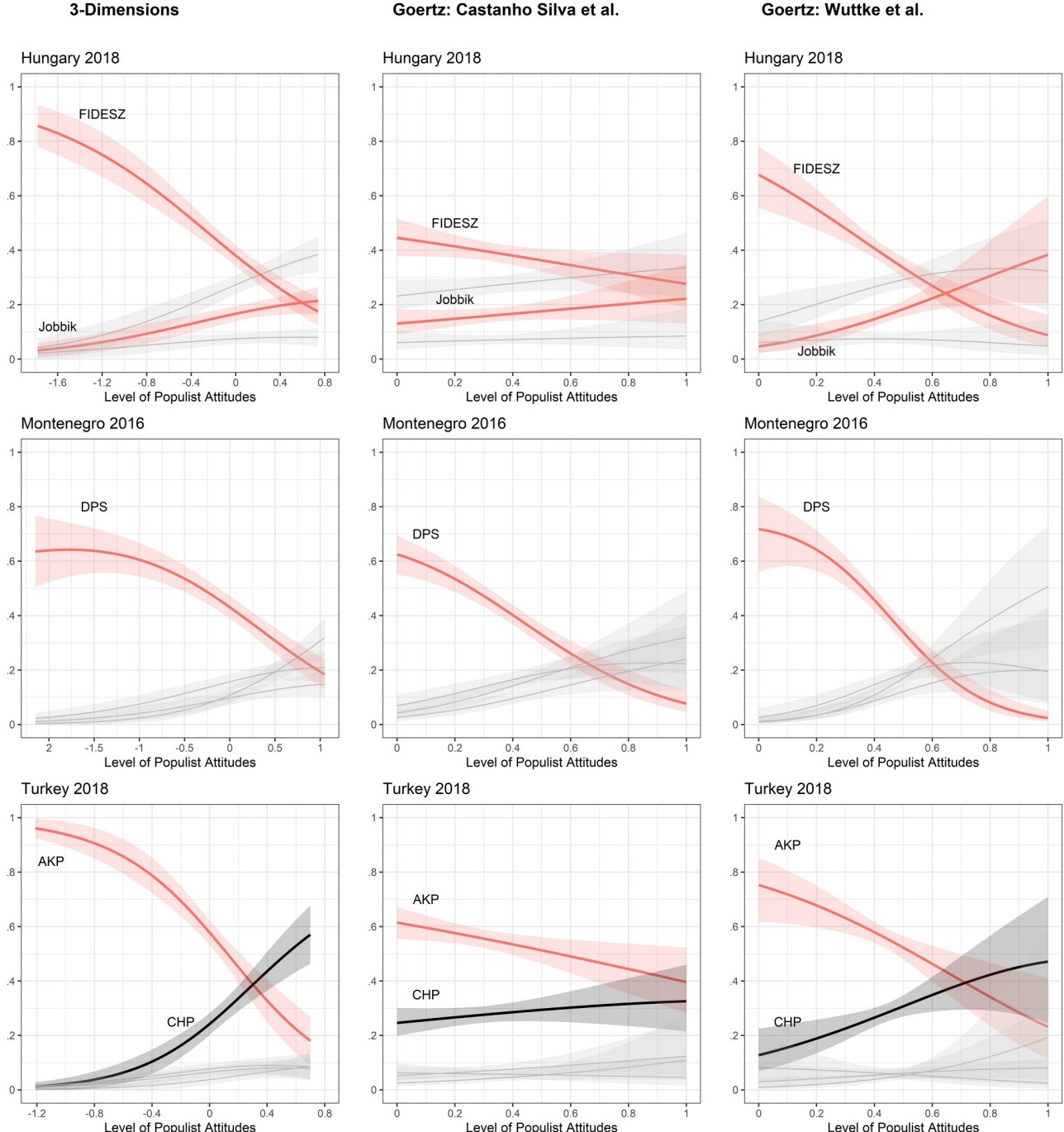

**Fig 4. Predictions of populist vote across countries (robustness checks).** Predictions of vote choice are based on multinomial structural equation models (3-Dimensions) and multinomial regression models (Goertz) with 95% confidence intervals.

as compared to the findings presented in Figs 2 and 3. The lone exception is Brazil where the Goertzian version of the Wuttke et al. items indicates a positive relationship with the PSL vote which has just been elected to power in the 2018 elections. The results from Hungary provide particularly strong evidence for our hypothesis that populist attitudes scales may fail to function when populist parties are in power. Here we find that the propensity to vote for the ruling

FIDESZ party decreases quite substantially as populist attitudes increase. However, higher populist attitudes increase the propensity to vote for Jobbik—a rival right-wing populist party which was not in power.

Finally, we carried out further robustness checks using the CSES items along with five other populist attitudes scales [1–3, 5, 6] using additional data from Greece (2016) and Japan (2019). In the analysis of the Greek data, we found only one scale developed by Stanley (2011) for which there was a positive, albeit very small, relationship between populist attitudes and vote choice for SYRIZA, the populist party which was elected to power in September 2015 (see S9 Fig in S1 File). All other scales indicate either no effect or a slightly negative effect. This is particularly interesting as the Greek data from the CSES in 2015 showed a positive relationship with voting for SYRIZA when the party was still in opposition. Thus, populist attitudes scales fail to predict support for populist parties in power regardless of which populist attitudes scale we use. As discussed above, Japan, unlike Greece, is a complex edge-case in terms of populism. Using original survey data conducted in 2019 we show that the negative relationship between LDP vote and populist attitudes also holds for the scales developed by Castanho Silva et al. and Schulz et al. (see S10 Fig and Appendix B in S1 File). Thus it also becomes difficult to study parties which have adopted some populist rhetoric.

## Discussion and conclusions

Overall, we conclude that populist attitudes scales, as they are currently operationalized, fail to predict vote choice for populist parties which are in power. In our eyes, this appears mainly be driven by the fact that all scales use items for the anti-elite dimension which over-specify the concept by focusing on negative aspects of parties and politicians. Although anti-establishment rhetoric of some form is a key selling point of populist parties around the world, its targets can be varied and flexible; a challenger party which has campaigned on an anti-government, anti-politician platform will likely refocus on other elites like the media, academics, bureaucrats or large corporations once they are in office, while a mainstream party that has only adopted aspects of populist rhetoric (perhaps to counteract the appeal of such a challenger party) will likely focus on non-political elites as its primary scapegoat from the outset.

One underlying reason for this problem with populist attitudes scales may be the somewhat homogeneous contexts in which the various scales used to study the demand side of populism have been developed. Unfortunately for cross-national research, the studies which led to the creation of these scales were often carried out in single countries and mostly in Western Europe—contexts in which populism was, until recently, almost exclusively the realm of small challenger parties. This homogeneity makes it especially important to test how populist attitudes scales work in other countries and political contexts. Our study is the first to date to investigate populist attitudes across multiple Asian countries and to provide direct comparison with countries in Europe and the Americas. This wide set of comparisons also allowed us to investigate the connection between populist demand and supply both in countries where populists are in opposition (as was the case in almost all places where these scales were developed) and in countries where populists are in government, either through having taken power as a challenger party, or through a mainstream party having adopted populist rhetoric for strategic purposes.

Existing measurements did not work to predict populist vote choice in countries where populist parties are in power. While arguments about the thermostatic nature of populist attitudes and the importance of there being a political aspect to populism must be taken into account, we argue that for most purposes, this constitutes a measurement error—the main reason for which lies in a conceptual problem with the construction of these scales, whereby

questions designed to measure "anti-elite" feelings are written in ways that solely concern negative feelings about political or governmental elites. This excludes the possibility of respondents' anti-elite attitudes instead being directed at other groups, either internal or external to their country—a factor which becomes especially pressing when populist parties are in power, as the election to office of their preferred populist leaders make citizens with populist attitudes, having "won" within the rules of the existing political system, less likely to agree with broadly anti-political or anti-governmental statements. As populist actors change their rhetoric to shift the empty signifiers of the "evil elite" from politicians to academics, journalists, judges, or minority groups, their supporters' perceptions also change. The perceived obstacle to the exercise of the popular will is no longer those in political office (now occupied by populists, whom they see as instruments of the popular will), but other conspiratorial forces in the media, academia, civil society or elsewhere. It is important to keep this shifting and multifarious nature of the concept of "eliteness" in mind when conducting this kind of research.

A new scale for measuring populist attitudes in future research could address this problem in two major ways: either by including various other elites in the scale, or by specifying the elite only vaguely, like many populists themselves do. Neither of these approaches provides a simple panacea, especially once the requirement for a populist attitudes scale to work cross-nationally is considered. Including questions measuring anti-elite sentiment towards other elite groups (business leaders, bureaucrats, academics, journalists, and so on) would have the advantage of allowing researchers to identify varieties of populism according to their targets, but would in the process create a very large survey battery whose inclusion in survey projects such as the CSES could be hard to justify. Moreover, the types of elites who resonate with citizens in different countries might be highly specific: we might think of anti-elite sentiment targeting groups such as LGBT activists in Eastern Europe, Chinese investors in developing countries in South-East Asia, or U.S. foreign policy actors in Latin America, none of which might be meaningful forms of anti-elitism to citizens in other regions. Using a vague specification of the elite and allowing respondents to assign their own meanings to the concept, however, raises other problems. The construction of the concept of "eliteness" may differ significant within countries, but is subject to even more difference cross-nationally, especially once the challenge of translating the term into other languages with its nuance intact is taken into consideration. Furthermore, there is also evidence that the activation of populist attitudes is subject to factors like corruption, failures of representation or economic and social crisis which can add to the existing complexity to gauge support for populist parties [70, 71]. Thus, significant further work is required on the development and refinement of survey batteries investigating citizens' populist attitudes, but finding ways to effectively allow for this complex and dynamic construction of anti-elite attitudes will be of significant value to comparative and cross-national studies.

## Supporting information

**S1 File.**
(PDF)

## Acknowledgments

We thank Bruno Castanho Silva for making the data available to us. We are also grateful to very helpful comments from Uwe Backes, Eric Chen-hua Yu, Charles Crabtree, Han Dorussen, Marc Helbling, Rob Johns, Steffen Kailitz, Axel Klein, Keiichi Kubo, Paul Marx, Maurits Meijers, Hannes Mosler, Christoph Nguyen, Saskia Ruth-Lovell, Gert Pickel, Sigrid Roßteutscher,

Jack Seddon, Atsushi Tago, Chung-min Tsai, Alexander Wuttke, and the participants of the NCCU-Waseda Research Colloquium Series, the WE-SPICE workshop between Waseda University and University of Essex, the PinEAD workshop at the IN-EAST of the University of Duisburg-Essen, and the workshop organized by the FGZ, IFRiS, and HAIT at the University of Leipzig in November 2019.

## Author Contributions

**Conceptualization:** Sebastian Jungkunz, Robert A. Fahey, Airo Hino.

**Data curation:** Sebastian Jungkunz.

**Formal analysis:** Sebastian Jungkunz.

**Funding acquisition:** Airo Hino.

**Investigation:** Sebastian Jungkunz.

**Methodology:** Sebastian Jungkunz.

**Project administration:** Sebastian Jungkunz, Robert A. Fahey, Airo Hino.

**Resources:** Sebastian Jungkunz.

**Software:** Sebastian Jungkunz.

**Supervision:** Sebastian Jungkunz, Robert A. Fahey, Airo Hino.

**Validation:** Sebastian Jungkunz.

**Visualization:** Sebastian Jungkunz.

**Writing – original draft:** Sebastian Jungkunz, Robert A. Fahey.

**Writing – review & editing:** Sebastian Jungkunz, Robert A. Fahey, Airo Hino.

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
