## [Decision Letter · Decision Letter 0]

6 Oct 2021

PONE-D-21-24026How populist attitude scales fail to capture support for populists in powerPLOS ONE

Dear Dr. Jungkunz,

Thank you for submitting your manuscript to PLOS ONE. After careful consideration, we feel that it has merit but does not fully meet PLOS ONE’s publication criteria as it currently stands. Therefore, we invite you to submit a revised version of the manuscript that addresses the points raised during the review process.

The two reviewers are quite positive and I tend to agree with them. They provide clear comments and suggestions, and I encourage you to provide a response letter to these evaluations. Also, please make sure to upload the data and your code, which should be accessible *during the review process.* 

We look forward to receiving your revised manuscript.

Kind regards,

Jean-François Daoust

Academic Editor

PLOS ONE

Journal Requirements:

Reviewers' comments:

Reviewer's Responses to Questions

**Comments to the Author**

1. Is the manuscript technically sound, and do the data support the conclusions?

Reviewer #1: Yes

Reviewer #2: Yes

2. Has the statistical analysis been performed appropriately and rigorously? 

Reviewer #1: Yes

Reviewer #2: Yes

3. Have the authors made all data underlying the findings in their manuscript fully available?

Reviewer #1: No

Reviewer #2: Yes

4. Is the manuscript presented in an intelligible fashion and written in standard English?

Reviewer #1: Yes

Reviewer #2: Yes

5. Review Comments to the Author

Reviewer #1: The study makes a straightforward argument about the varying role of populist attitudes for predicting vote choice for populist parties. In one of the first publications using the CSES populist attitudes scale, this study reveals the surprising finding that in some countries citizens with higher levels of populist attitudes are less likely (and not more likely, as one would expect) to vote for a populist party. This finding is notable in itself. Moreover, the study also explains this finding, arguing that current scales of populist attitudes are highly sensitive to whether a populist party is in government or not. Particularly, the authors argue that – due to the item wording – populist attitudes scales lose explanatory power when a populist party is in government because these scales only tap into attitudes towards politicians rather than societal elites as a whole. This argument is intuitively convincing and the finding has ramifications for the validity of populist attitudes scales and future research on populist attitudes. I have reviewed a previous version of this paper a few months ago and congratulate the authors on the various improvements of an already promising draft. While I had submitted several critical comments on the earlier version of this article, this time around I have only minor suggestions.

Even though I think that the main observation (how voters react to populists in power) raises many questions that are worth thinking about, I see the contribution of this manuscript not so much as trying to enlighten these questions in a substantive matter. Rather, the contribution is practical and methodological, advising scholars of populist attitudes that their scales work differently depending on whether populists are in power. In doing so, this paper is likely more than the beginning (even though similar work already exists, eg. Castanho Silva 2019) than the end of this line of research and the study helps other populism researchers avoid consequential mistakes.

Still, my most significant reservation concerns this substantive dimension of the observed phenomena: The authors interpret the observed and indeed very interesting phenomena of varying associations between populist parties and populist attitudes across countries largely as evidence of measurement error: Populist citizens are still considered populist but under certain conditions the scale no longer achieves to capture these sentiments. Yet, upon closer inspection there is not much evidence given for this interpretation in contrast to other conceivable interpretations: It is also possible that previously populist citizens have simply changed their minds after their preferred populist parties took power and they are no longer populist. Many authors consider populism as a redemptive force and maybe populists have now gotten what they wanted, so there is no longer the need for populist attitudes (populism as a thermostatic attitude). Answering this question requires engaging with conceptual questions on the exact definition of what anti-elitism means and which types of actors may be considered here (eg political or non-political actors – to me it is not self-evident that populism must necessarily also include non-political actors as it is a politics-focused concept). While there is no need to engage with these questions in the empirical section of the article, I would urge the authors to at least contemplate these questions and discuss them either as limitations of their study or as potential avenues for further research. Primarily, I would ask the authors to consider on what evidential or conceptual bases they think the conclusion is warranted that we are dealing with a measurement problem as opposed to actual changes in attitudes.

In the following I will mention several minor aspects in no particular order:

- Authors use very recent data that was published only a few months ago, making the study very much up to date.

- Study exhibits careful scrutiny of the measurement instrument and the available data. Indeed, there is no straightforward operationalization of the CSES populism scale. Hence, I think it is laudable and useful that the authors transparently discuss the various problems they faced and decisions they took when aggregating the items to an index of populist attitudes. In this vein, it also seems sensible to present alternative specifications of how populist attitudes can be captured using the CSES data. (I also want to add that the 3-dimensional structure the authors propose seems very reasonable)

- The authors conduct CFA to test whether the data fit populist attitudes as described by Wuttke et al. But Wuttke et al. conceive of populist attitudes as non-compensatory which does not imply a correlation between the subcomponents of populism (even though a strong correlation between the indicators of each subcomponent is expected). But the presented CFA likely specified such a correlation between the subdimensions and the strength of that correlations determine the reported goodness of fit indices – if I am not mistaken. Hence, I am wondering whether the model and the conclusion fit the concept specification in this case.

- I was first wondering whether it is a reasonable choice to conduct a multinomial (instead of a binary) regression, but it is an excellent choice as proven with the very helpful and informative Figure 1!

- The manuscript states that “we coded populist parties along the schema provided 295

by the PopuList [50] except for borderline cases.” Which were these borderline cases, why have you decided not to follow PopuList in these cases and how does this change the substantive conclusions drawn from the empirical analysis?

- On page 16, it reads that the second column of Table 1 explains the three dimensions of populist attitudes, but that column gives only the question wordings.

- Please include the figures in the next version of the manuscript right in the text as the manuscript is otherwise hard to navigate. More importantly, the figures are extremely pixelated right now

- The replication data are (not yet?) uploaded on the linked dataverse

I hope that these comments were helpful to the authors and wish them good luck in further pursuing this project.

Sincerely,

Alexander Wuttke

Reviewer #2: # PONE-D-21-24026

This paper argues that the scales used to measure populist attitudes have a different substantive meaning in different political contexts. With multiple operationalization of populist attitude scales using CSES Module 5 data, the authors show that while populist attitudes predict support populist parties in countries were populist parties are in opposition, the predictive power of populist attitudes for the populist vote is not present in countries in which the populist party is in office.

I think this paper is well written and methodologically well done. Its analysis and its main findings are an important contribution to the study of populism on the individual level. The finding that the effect of populist attitudes on populist voting depends on the incumbency context in each country is an important finding. As such I think it is fit to be published in PlosOne. Nevertheless, I have a few minor suggestions on how to make the paper stronger.

## Populist Attitudes and Populist Voting

The paper convincingly shows how the populist attitude scales behave differently in different political contexts. To this end, the authors show that the predictive quality of populist attitudes for populist voting differs when populist party is in office. While I agree that this is a good way to get a grasp of the differential substantive meaning of populist attitude scales, I do not agree that the central aim of the concept of populist attitudes is that it predicts populist voting, as the authors suggest on p. 6. Rather, populist attitude scales have a much broader application. Populism (and populist attitudes) can be considered to be a view on political representation (Urbinati 2019). This is not to say that populist attitudes haven’t been used or validated by measuring their effect on populist voting. It is simply that the concept’s utility is broader. I think the paper would be stronger if it reflects on this issue a bit.

## Variations of Populist Attitudes Scales.

Although it is beyond the author’s discretion, it is a shame that the CSES failed to include the original six items as proposed by Akkerman et al. Instead the authors have to rely on the other items included in the CSES. As the authors also acknowledge, the item E3004_1 on compromise does not fit the Manichean category. A Manichean world view implicates that politics is a struggle between good and evil. While both the rejection of compromise and the Manichean worldview have a anti-pluralist component, the Manichean worldview sees the struggle between opposing camps as an inherently moral one. As such, I would suggest that E3004_1 should be labelled as measuring Manichean worldview, even if Castanho Silva et al (2020) proposed this. I agree with Wuttke et al (2020) that it rather fits a representation-related dimension. Interestingly, also the original AKkerman et al. scale failed to find a good item to capture the Manichean worldview.

In the Wuttke et al. (2020) CSES scale, an item about corruption by politicians is included. I believe that corruption perceptions cannot be equated or subsumed under ‘populist attitudes’. As Meijers and Zaslove argue in their critique of the Chapel Hill Expert Survey measuring populism (also) as party positioning on corruption, populists do not aim to fight actual corruption (i.e. misuse of power for private gain). Rather, as Taggart (2018) argues, participating in politics is corruptING. As such, I would remove the item from the main analysis. This might affect the results as Goertzian approach proposes to use the minimum score for all dimensions/components. The full CSES scale proposed by Wuttke et al can be shown in the appendix.

A strength of the paper is that it replicates the analysis for Greece and Japan by testing different types of scales including the original Akkerman et al (2014) scale and the Schulz et al (2017) scale. Yet, I think the paper would benefit from a more in-depth discussion on how these scales differ (and, importantly, are similar) in the “Data & Operationalization” section. Currently, this is only briefly discussed at the end.

## Discussion: What Now?

While I understand that it is beyond the scope of this paper to develop and test a new scale of populist attitudes which circumvents the problems diagnosed with current populism scales, I think the authors could do more to discuss what such a new populist scale in future research should look like. The authors’ main argument is that currently the populist attitudes scales prime respondents to think about political elites only. Yet, how could this issue be addressed? Removing all references to politicians and only refer to elites would render the questions perhaps too vague and too complex. Would everyone understand what (and who) elites are? It would be great if the paper could propose ways in which one could measure populist attitudes in cross-national fashion independent of the political context.

## References

Akkerman A, Mudde C, Zaslove A. How Populist Are the People? Measuring Populist Attitudes in Voters. Comparative Political Studies. 2014;47(9):1324–1353.

Castanho Silva B, Jungkunz S, Helbling M, Littvay L. An Empirical Comparison of Seven Populist Attitudes Scales. Political Research Quarterly. 2020;73(2):409–424.

Taggart, P. (2018). Populism and ‘unpolitics.’ In G. Fitzi, J. Mackert, & B. Turner (Eds.), Populism and the Crisis of Democracy: Volume 1: Concepts and Theory (pp. 79–87). Routledge.

Meijers, Maurits J., and Andrej Zaslove. 2021. “Measuring Populism in Political Parties: Appraisal of a New Approach.” Comparative Political Studies 54 (2): 372–407.

Schulz A, Müller P, Schemer C, Wirz DS, Wettstein M, Wirth W. Measuring Populist Attitudes on Three Dimensions. International Journal of Public Opinion Research. 2018;30(2):316–326.

Urbinati, Nadia. (2019) Me the People. Harvard University Press.

Wuttke A, Schimpf C, Schoen H. When the Whole is Greater Than the Sum of its Parts: On the Conceptualization and Measurement of Populist Attitudes and Other Multi-dimensional Constructs. American Political Science Review. 2020;

6. PLOS authors have the option to publish the peer review history of their article (what does this mean?). If published, this will include your full peer review and any attached files.

Reviewer #1: **Yes: **Alexander Wuttke

Reviewer #2: **Yes: **Maurits Meijers

---

## [Author Response · Author response to Decision Letter 0]

20 Nov 2021

We provided a separate response letter as part of the submission files. We also uploaded all replication materials to a Harvard Dataverse which is cited in the paper.

Thank you very much.

---

## [Decision Letter · Decision Letter 1]

9 Dec 2021

How populist attitudes scales fail to capture support for populists in power

PONE-D-21-24026R1

Dear Dr. Jungkunz,

We’re pleased to inform you that your manuscript has been judged scientifically suitable for publication and will be formally accepted for publication once it meets all outstanding technical requirements.

Kind regards,

Jean-François Daoust

Academic Editor

PLOS ONE

Additional Editor Comments (optional):

Reviewers' comments:

Reviewer's Responses to Questions

**Comments to the Author**

1. If the authors have adequately addressed your comments raised in a previous round of review and you feel that this manuscript is now acceptable for publication, you may indicate that here to bypass the “Comments to the Author” section, enter your conflict of interest statement in the “Confidential to Editor” section, and submit your "Accept" recommendation.

Reviewer #1: All comments have been addressed

Reviewer #2: All comments have been addressed

2. Is the manuscript technically sound, and do the data support the conclusions?

Reviewer #1: Yes

Reviewer #2: Yes

3. Has the statistical analysis been performed appropriately and rigorously? 

Reviewer #1: Yes

Reviewer #2: Yes

4. Have the authors made all data underlying the findings in their manuscript fully available?

Reviewer #1: Yes

Reviewer #2: Yes

5. Is the manuscript presented in an intelligible fashion and written in standard English?

Reviewer #1: Yes

Reviewer #2: Yes

6. Review Comments to the Author

Reviewer #1: The authors have satisfactorily addressed all concerns and I congratulate the authors on their interesting study.

Sincerely,

Alexander Wuttke

Reviewer #2: Dear authors,

Many thanks for these revisions. I think you have addressed all of my comments, which were very minor to start with, very well.

7. PLOS authors have the option to publish the peer review history of their article (what does this mean?). If published, this will include your full peer review and any attached files.

Reviewer #1: **Yes: **Alexander Wuttke

Reviewer #2: **Yes: **Maurits Meijers

---

## [Editor Report · Acceptance letter]

23 Dec 2021

PONE-D-21-24026R1 

How populist attitudes scales fail to capture support for populists in power 

Dear Dr. Jungkunz:

I'm pleased to inform you that your manuscript has been deemed suitable for publication in PLOS ONE. Congratulations! Your manuscript is now with our production department. 

Kind regards, 

on behalf of

Dr. Jean-François Daoust 

Academic Editor

PLOS ONE